# Role of Abscisic Acid, Reactive Oxygen Species, and Ca^2+^ Signaling in Hydrotropism—Drought Avoidance-Associated Response of Roots

**DOI:** 10.3390/plants13091220

**Published:** 2024-04-28

**Authors:** Baris Uzilday, Kaori Takahashi, Akie Kobayashi, Rengin Ozgur Uzilday, Nobuharu Fujii, Hideyuki Takahashi, Ismail Turkan

**Affiliations:** 1Department of Biology, Faculty of Science, Ege University, Bornova 35100, Izmir, Turkey; 2Graduate School of Life Sciences, Tohoku University, Katahira, Sendai 980-8577, Japan; 3Research Center for Space Agriculture and Horticulture, Graduate School of Horticulture, Chiba University, Matsudo, Chiba 271-8510, Japan; 4Faculty of Agricultural Sciences and Technologies, Yasar University, University Street, No. 37-39, Bornova 35100, Izmir, Turkey

**Keywords:** abscisic acid (ABA), Ca^2+^ signaling, drought stress, MIZ1, GNOM/MIZ2, reactive oxygen species (ROS), root hydrotropism

## Abstract

Plant roots exert hydrotropism in response to moisture gradients to avoid drought stress. The regulatory mechanism underlying hydrotropism involves novel regulators such as MIZ1 and GNOM/MIZ2 as well as abscisic acid (ABA), reactive oxygen species (ROS), and Ca^2+^ signaling. ABA, ROS, and Ca^2+^ signaling are also involved in plant responses to drought stress. Although the mechanism of moisture gradient perception remains largely unknown, the sensory apparatus has been reported to reside in the root elongation zone rather than in the root cap. In Arabidopsis roots, hydrotropism is mediated by the action of MIZ1 and ABA in the cortex of the elongation zone, the accumulation of ROS at the root curvature, and the variation in the cytosolic Ca^2+^ concentration in the entire root tip including the root cap and stele of the elongation zone. Moreover, root exposure to moisture gradients has been proposed to cause asymmetric ABA distribution or Ca^2+^ signaling, leading to the induction of the hydrotropic response. A comprehensive and detailed analysis of hydrotropism regulators and their signaling network in relation to the tissues required for their function is apparently crucial for understanding the mechanisms unique to root hydrotropism. Here, referring to studies on plant responses to drought stress, we summarize the recent findings relating to the role of ABA, ROS, and Ca^2+^ signaling in hydrotropism, discuss their functional sites and plausible networks, and raise some questions that need to be answered in future studies.

## 1. Introduction

Being sessile, plants have evolved a wide spectrum of response mechanisms and adaptations to cope with different environmental stresses. Drought is considered one of the most challenging environmental stresses, causing up to 25% of the losses in crop yield worldwide and threatening the global food supply [1]. According to the predictions of the International Water Management Institute (IWMI), one-third of the world population will be living in regions facing severe water scarcity by 2025 (IWMI Water Brief 1; https://pdf.usaid.gov/pdf_docs/Pnach595.pdf accessed on 26 March 2024). Because plants may experience too little or too much water with the accelerating effects of global climate change in the next coming decades, the control of water uptake and use in plants is extremely important [2]. Therefore, to improve the crop yield worldwide, understanding the characteristics of root growth and development under water scarcity has emerged as one of the most important tasks for plant biologists.

The loss of water from cells and the accessibility to available water are stresses that were likely imposed very early in evolution in terrestrial environments, where water was the main constraint. Hence, to maintain a favorable water balance and turgidity under water-limiting conditions, plants have developed four adaptive mechanisms: (1) drought avoidance, (2) drought tolerance, (3) drought escape, and (4) drought recovery. Among these, drought avoidance and drought tolerance are the two major mechanisms used by plants to achieve drought resistance [3].

Drought avoidance strategies aim to (1) reduce water loss by inducing rapid stomatal closure, decreasing leaf area by leaf rolling, and increasing wax accumulation on the leaf surface; and (2) enhance water storage in specific organs capable of forming fleshy water-storing tissues [4]. Moreover, to adapt to dry conditions, plants display considerable developmental plasticity in their root systems, which evolved in the sporophytes of early vascular plants during their initial expansion on land in the Early Devonian period (395–410 million years ago). The adaptations and responses of the plant root system to the variations in water availability are considered an excellent example of a drought avoidance strategy [5].

Generally, to enhance water uptake and sustain a higher water potential under water-limiting conditions, plants increase their root depth, volume, and distribution, which are mainly influenced by the depth and range of soil moisture [6]. Hence, a well-developed root system, characterized by an increased root depth, root density, and root/shoot ratio, is essential for accomplishing drought avoidance. Besides their involvement in the traits related to drought tolerance, abscisic acid (ABA), reactive oxygen species (ROS), and calcium (Ca^2+^) play roles in root growth and root system architecture under drought stress [7].

Even under normal conditions, the soil water availability affects the growth of the root system as a whole. In general, the water content varies among the different areas of soil, with some areas having more water than others [8,9,10], and plant roots respond to this variation in soil water content by altering their growth rate and direction. The ability of plants to direct their roots to grow toward water in the soil is called hydrotropism, a directed root growth response described in the early 19th century [11]. Hydrotropism involves root bending, owing to differential growth, and guides the roots towards areas with a higher water potential in a heterogenous soil environment.

The mechanisms underlying root hydrotropism have been investigated in recent years, and advances in genomic tools as well as the increased availability of mutants have provided valuable insights into some of the important aspects of the hydrotropic response, which include the perception and signal transduction of moisture gradients. One of the major breakthroughs in the field of root hydrotropism was achieved using *Arabidopsis thaliana* mutants defective in ABA synthesis or signaling; these mutants exhibited a reduced rate of the hydrotropic response, indicating that ABA signaling is essential for hydrotropism [12]. Approximately 14 years later, Krieger et al. [13] reported that ROS mediate two different directed growth responses of roots, i.e., gravitropism and hydrotropism, and demonstrated the role of hydrogen peroxide (H_2_O_2_) as a secondary messenger in hydrotropism. Additionally, changes in the free Ca^2+^ concentration in the cytosol have also been implicated in root hydrotropism [14,15,16]. All of these findings signify an interplay between ABA, ROS, and [Ca^2+^] fluxes during hydrotropism, a drought avoidance response in plants. Although ROS signaling seems to be linked to ABA and [Ca^2+^] fluxes in general, the investigation of the role of ROS and ROS-mediated signaling in the hydrotropic response of roots is of critical importance. The cells and/or tissues involved in these signaling processes and the apparatus involved in sensing the moisture gradients (hydrostimulation) also need to be clarified.

Throughout this review, we evaluate the roles of ABA, ROS, and Ca^2+^ signaling in the hydrotropic response of different plant species. We also present new perspectives and discuss the questions that need to be answered in the future to better understand the mechanisms of root hydrotropism.

## 2. Water Deficiency Perception, ABA Core Signaling Activation, and Their Relationship with Root Hydrotropism

To activate signaling networks and initiate suitable transcriptional regulation in response to water-deficient conditions, plants must first perceive the deficiency in soil water in the roots. However, no specific drought stress receptors have been reported in plants to date. In yeast, studies on the osmotic stress response revealed the high osmolarity glycerol (HOG) pathway, which relies on the histidine kinase Sln1, an osmosensor that monitors changes in the cell turgor pressure [17]. HOG activates a cascade of phosphorylation events via mitogen-activated protein kinases (MAPKs), which consequently induce the transcription of glycerol biosynthesis genes [18]. A similar signal perception pathway also exists in plants [19], as indicated by studies on *Arabidopsis thaliana*. The *AtHK1* gene, which is homologous to *Sln1*, was expressed in Arabidopsis roots under osmotic stress [20]. Transgenic Arabidopsis plants overexpressing *AtHK1* were drought tolerant, whereas *athk1* mutants were more sensitive to drought [21]. It has been suggested that AtHK1 positively regulates ABA signaling rather than acting as an osmosensor, because ABA sensitivity was affected in these plants. This implies that the osmosensing pathway in Arabidopsis is different from that in yeast. The role of *AtHK1* in root growth regulation was also reported in poplar [22], where the *AtHK1* protein activated a phosphorelay pathway potentially involved in osmotic stress sensing [23]. In our study, no difference could be detected between the hydrotropic response of the *athk1* mutant roots and that of the wild-type roots (Hiroki Takahashi et al., unpublished data). A question therefore arises regarding the role of *AtHK1* in the hydrotropic response of Arabidopsis roots.

An essential part of drought perception by roots is regulated by plant hormones, predominantly ABA, and the biosynthesis of ABA in roots is well documented as an early response to water deficits [24]. The level of ABA in plants during vegetative growth is tightly controlled by a balance between its biosynthesis and catabolism to mediate a response to environmental stresses such as drought, high salt, and low temperatures, all of which impose cellular osmotic stress [25,26]. ABA regulates the expression of various stress-responsive genes including those encoding LEA proteins, dehydrins, and other protective proteins [27], and induces processes involved in the maintenance of cellular turgor and the synthesis of osmoprotectants and antioxidant enzymes [28].

ABA acts as a long-distance signal under water-limiting conditions, conveying the onset of stress from roots to shoots, which eventually results in water-saving responses such as stomatal closure and reduced leaf expansion, both of which are considered drought avoidance responses [29]. ABA is also involved in robust root growth and other architectural modifications under drought stress [30]. Root growth parameters, such as length, weight, volume, and density, increase substantially under drought stress, especially in dry areas, to maintain sufficient water potential in plant tissues. Hence, at the whole-plant level, ABA levels are slightly elevated under mild water stress to maintain root growth but inhibit shoot growth, leading to an increased root-to-shoot ratio, which has been used as a criterion for drought resistance [31,32]. On the other hand, high levels of ABA can inhibit root growth to conserve energy for the induction of defense mechanisms. This suggests that ABA levels act as a switch to transition from drought avoidance to drought tolerance, as shown in Figure 1; Arabidopsis plants display robust root growth and low ABA levels under mild drought stress and reduced root growth and high ABA levels under severe drought stress [33,34].

Thus, the above-mentioned drought avoidance responses, including stomatal closure, reduced leaf area, and robust root growth, are regulated by the ABA core signaling pathway, which comprises the following three factors involved in the perception and relay of the ABA signal: PYR/PYL/RCAR family proteins (ABA receptor), type 2C protein phosphatases (PP2Cs; negative regulator), and class III SNF1-related protein kinases 2 (SnRK2s; positive regulator) [35]. The ABA core signaling cascade is initiated when ABA binds to PYR/PYL/RCAR proteins. This results in the inhibition of PP2Cs through the formation of the ABA–PYR/PYL/RCAR–PP2Cs complexes, which activates downstream ABA signaling, i.e., the accumulation of phosphorylated SnRK2s, phosphorylation of ABA-responsive element binding factors, and ABA-related gene expression for appropriate cellular responses [36]. Hence, SnRK2s play a positive regulatory role in the drought tolerance of Arabidopsis roots. Accordingly, the transgenic Arabidopsis plants expressing *35S::SnRK2C-GFP* displayed higher overall drought tolerance than the control plants; ABA activated the SnRK2 member AAPK in the guard cells of *Vicia faba* leaves, causing stomatal closing; and the *Arabidopsis snrk2.6* (*ost1*) mutant was defective in ABA-induced stomatal closure [37,38].

Using the ABA biosynthesis mutant *aba1-1*, ABA signal transduction mutant *abi2-1*, and a sextuple receptor mutant, Takahashi et al. [12] and Antoni et al. [35] conceivably demonstrated that, as in the case of stomatal closure, the process of hydrotropism also requires core ABA signaling in Arabidopsis. Mutants defective in ABA biosynthesis or signaling exhibited a reduced rate of hydrotropic response. During the hydrotropic response, Arabidopsis SnRK2 protein kinases function as positive regulators of the ABA signaling pathway [39,40]. The *snrk2.2 snrk2.3* double mutant showed ABA-insensitive root growth phenotypes [41], indicating that both *SnRK2.2* and *SnRK2.3* are involved in ABA signaling and are regulated by ABA. Additionally, the *snrk2.2 snrk2.3 snrk2.6* triple mutant showed severe phenotypes consistent with impairments in ABA signaling and water stress responses [39,42]. Moreover, the *snrk2.2 snrk2.3* double mutant exhibited a reduced hydrotropic response [43].

The above studies suggest that ABA is a regulator responsible for root hydrotropism, which is accompanied by differential osmotic stress to the roots in the presence of moisture gradients. A mathematical model suggested that the asymmetrical redistribution of ABA likely occurs between the convex and concave sides of the elongation zone of Arabidopsis roots in response to moisture gradients [43]. Because ABA maintains a higher rate of cell expansion under water stress conditions [31,32], the gradient of ABA could be responsible for differential root growth during the hydrotropic response. In addition, the application of exogenous ABA could reverse the decreased hydrotropic response of the roots in the *aba1-1* mutant, as mentioned before. We presume that exogenous ABA perturbs the formation of an ABA gradient across the root. Accordingly, the role of an ABA gradient in the hydrotropic response needs further investigation. On the other hand, the receptor turnover is an important aspect of hormone signaling, which is also true for ABA receptors on the membranes. Previously, the RING-type E3 ubiquitin ligase RSL1 was shown to interact with the ABA receptors PYL4 and PYR1 at the plasma membrane [44]. Additionally, *RSL1* overexpression lines exhibited reduced ABA sensitivity, whereas *rsl1* RNA interference (RNAi) lines showed enhanced sensitivity to ABA [44]. Similarly, a recent study showed that ALG2-INTERACTING PROTEIN-X (ALIX), an endosomal sorting complex required for transport (ESCRT-III)-associated protein, directly binds to and degrades ABA receptors in late endosomes. While ABA hypersensitivity was reported in the *alix-1* knockdown mutant, *alix-1* plants with an impaired PYR/PYL/RCAR receptor did not exhibit this phenotype [45]. Interestingly, both ionic and nonionic hyperosmotic treatments induced clathrin-mediated endocytosis and simultaneously attenuated exocytosis in root meristematic cells [46]. Moreover, hydrotropic stimulation or exogenous ABA application induce PLDζ2 expression in the root tip, and *pldζ2* plants showed delays in gravitropic and hydrotropic responses [47], which suggests that phosphatidic acid-dependent endocytosis is involved in root tropisms. PLDζ2-induced clathrin-mediated endocytosis also plays a role in the internalization and degradation of the auxin efflux carrier PIN2 [48]. Whether the changes in ABA receptor turnover on either side of the root are similar to those in the PIN2 turnover during hydrotropism deserves further investigation.

Kobayashi et al. [49] and Miyazawa et al. [50] identified *MIZU-KUSSEI 1* (*MIZ1*) and *GNOM/MIZ2*, respectively, both of which are essential for hydrotropism in Arabidopsis roots. *MIZ1* encodes a protein with an unknown function, which is conserved in the genome of terrestrial plants but not that of algae [49], and *GNOM*/*MIZ2* encodes an ADP-ribosylation factor-type G (ARF-GEF) protein that regulates vesicle trafficking [50]. The roots of both *miz1* and *miz2* mutants are impaired in the hydrotropic response but exhibit a normal gravitropic response. Thus, the functions of MIZ1 and GNOM/MIZ2 are unique to hydrotropism, and an analysis of these proteins will likely lead to the discovery of a novel mechanism underlying root hydrotropism. Importantly, exogenous ABA application upregulated *MIZ1* expression in Arabidopsis roots [51]. Although *MIZ1* was expressed in a broad range of tissues, including the root cap, cortex, and epidermal layers, in the root apical meristem and elongation zone, the cortex-specific expression of *MIZ1* and *SnRK2.2* in the *miz1* single mutant and *srnk2.2 srnk2.3* double mutant, respectively, could rescue the hydrotropic response of roots [43]. These results suggest that the differential growth of the cortex is a driving force of hydrotropism. Although the molecular crosstalk between MIZ1 and ABA remains an open question, the results led us to assume that ABA biosynthesis and signaling activated under drought conditions control *MIZ1* expression for inducing root hydrotropism. Also, one could postulate that an asymmetric increase in the ABA level under moisture gradients results in differential *MIZ1* expression in the root elongation zone. This possibility needs to be studied, because differential MIZ1 expression in the hydrotropically responding roots has not been observed [52]. The importance of cortex-controlled root hydrotropism is also supported by the fact that the removal or ablation of the root cap disrupted gravitropism but not hydrotropism [43], as discussed below in Section 4. Thus, it is possible that the apparatus for sensing moisture gradients exists in the root cortex of elongation zone.

## 3. Does ROS Signaling Participate in Root Hydrotropism?

A growing body of evidence shows that the generation of ROS is one of the most common plant responses to different environmental cues, acting as a hub at which various signaling pathways converge [53]. The rapid production of ROS is a common plant response to biotic and abiotic stresses, and thus is considered a basis for unifying signaling events [53,54]. ROS production has also been observed previously during normal root growth and during root tropic responses in maize and Arabidopsis [13,55].

ROS have been implicated in both the environmental responses and related developmental processes in roots, and the roles of the respiratory burst oxidase homologs RBOHC, RBOHD, and RBOHF have been established. The *RBOH* homologs encode plasma membrane-localized NADPH oxidases, which are responsible for the production of apoplastic O_2_^−^. O_2_^−^ is short-lived and is spontaneously or enzymatically (by superoxide dismutase) converted to hydrogen peroxide (H_2_O_2_), which mediates further ROS signaling. According to Krieger at al. [13], the directed growth of roots towards water is negatively regulated by RBOHC activity. Assays exposing the *rbohc* mutant to a low water potential demonstrated that the hydrotropic response of the *rbohc* roots was more pronounced than that of the wild-type roots. The enhanced hydrotropic bending in the *rbohc* roots was attributed to decreased H_2_O_2_ levels in the elongation zone. The *rbohd* mutant with lower H_2_O_2_ levels in whole seedlings but wild-type H_2_O_2_ levels in the root apices showed a comparable hydrotropic response to wild-type plants. These results indicate that RBOHC produces O_2_^−^, which is subsequently converted into H_2_O_2_, and that the elevated H_2_O_2_ level reduces hydrotropic bending. Additionally, the transcript abundance of *RBOHC* was greater than that of *RBOHD* and *RBOHF* in the elongation zone (Figure 2).

Although an asymmetric ROS distribution was detected between the convex and concave sides of the root elongation zone during gravitropism, this phenomenon was not observed in the distal or central elongation zone during hydrotropism [13]. It should be noted that the mentioned study [13] used a rhodamine-based fluorescent probe (dihydrorhodamine-123 [DHR]), which is the most sensitive to H_2_O_2_ [58] and is generally used to monitor cytosolic ROS levels [59]. Therefore, assays with DHR might not reflect the level of apoplastic ROS, which is generally produced by NADPH oxidases in response to environmental stimuli. On the other hand, analysis using diphenylene iodonium (DPI), a NADPH oxidase inhibitor, revealed that the root curvature increases during hydrotropism and decreases during gravitropism in the DPI-treated roots [13]. Moreover, the treatment of roots with an antioxidant such as ascorbate increases the curvature in hydrostimulated roots but, in turn, inhibits gravitropism. These findings suggest that NADPH oxidases and/or ROS do not play a role in the hydrotropic bending of Arabidopsis roots. Moreover, the root curvature accelerated by the modulation of ROS levels can be explained by an inhibition of the response to gravistimulation, which would enable the hydrostimulated roots to overcome gravitropism.

It has been suggested that autophagy is involved in the degradation of amyloplasts in the columella cells of hydrotropically stimulated roots, which could reduce gravitropism and thereby enhance hydrotropism [60,61,62]. Amyloplast degradation due to hydrotropic stimulation was observed in hydrotropic mutants such as *miz1*, *nhr1*, and *ahr1*, but it did not alter their hydrotropic response [62]. This result supports the idea that reduced gravitropism enhances the hydrotropic response; the ahydrotropic or altered hydrotropic responses of these mutants are independent of their gravitropism-interfered response. Interestingly, however, in plants expressing GFP-ATG8a, autophagosomes accumulated in the root curvature 2 h after hydrostimulation. Remarkably, several *atg* mutants showed an impaired hydrotropic response, indicating that autophagy is required for root bending [62]. A H_2_O_2_ biosensor (Hyper) showed that H_2_O_2_ accumulated in the root curvature at a similar rate as the autophagosomes during the hydrotropic response. The inhibition of the peroxidase and RBOH activities affected the root curvature either negatively or positively, indicating that low levels of ROS accelerate the hydrotropic response. Jiménez-Nopala et al. [62] hypothesized that ROS’ regulation of autophagy is involved in cell-wall-mediated differential growth in the root elongation zone and is required for the induction of hydrotropism. However, it should be noted that the chemical inhibition of RBOH decreases the root growth rate, and Hyper is a cytosolic H_2_O_2_ reporter that does not fully represent the apoplastic ROS levels.

## 4. How Does Ca^2+^ Signaling Function in Root Hydrotropism?

The evolutionarily conserved phenomenon of increased cytosolic Ca^2+^ in response to diverse environmental signals puts Ca^2+^ among the most important second messengers [63]. A low Ca^2+^ level (nanomolar levels) in the cytoplasm and high free Ca^2+^ level (millimolar levels) in the endoplasmic reticulum (ER), vacuoles, other organelles, and cell walls are tightly regulated by a system of cellular membrane-localized pumps, channels, or transporters. Hence, the sensitivity of cells to various environmental stimuli is dependent on their ability to sequester and utilize Ca^2+^ from internal stores [64]. Ca^2+^-sensing mechanisms that maintain specificity in terms of signal transmission and response generation have been an intense area of research recently. The “calcium signature”, the kinetics and magnitude of the Ca^2+^ concentration, varies with signals and possibly contributes to the specificity of responses [64]. Ca^2+^-sensing and -binding proteins further specify and amplify these signals and are involved in signal transmission and relay [65]. Some of these proteins with enzymatic activity can amplify the signals on their own, whereas other proteins with no enzymatic activity amplify and relay the signals by interacting with other proteins possessing enzymatic or transcriptional activity. Plant Ca^2+^ sensor proteins either act as sensor responders or sensor relay proteins [66]. Sensor responder proteins such as Ca^2+^-dependent protein kinases (CDPKs) perform both sensing and response functions. On the other hand, sensor relay proteins such as calcineurin B-like proteins (CBLs) and CBL-interacting protein kinases (CIPKs) bind effectively to Ca^2+^ ions, undergo conformational changes, and then interact with specific downstream proteins with amplification and relay functions usually via phosphorylation and dephosphorylation [67].

Calreticulin (CRT), a low-affinity high-capacity Ca^2+^-binding protein that exists in the ER lumen, acts as a calcium-binding peptide (CBP) to specifically increase Ca^2+^ storage in the nucleus and ER. Additionally, CRT also affects intracellular Ca^2+^ homoeostasis by modulating the storage, release, and transport of ER-localized Ca^2+^ [67]. Calcineurin, a Ca^2+^-calmodulin (CaM)-dependent serine–threonine phosphatase, has been implicated in various signaling events in cells. Accordingly, calcineurin modulates cellular processes in response to the secondary messenger Ca^2+^.

Tsou et al. [68] reported that plants expressing the CBP domain of CRT exhibited better survival under drought or salinity stress due to increased Ca^2+^ levels and exhibited greater root growth than the wild-type. *CIPK6*, a member of the *CIPK* gene family, was induced in CBP-expressing plants even under normal conditions. The loss of function of *CIPK6* abolished the stress tolerance of CBP-overexpressing transgenic plants. This demonstrates that Ca^2+^ stored in the ER can directly participate in signal transduction to alter gene expression.

It has been suggested that Ca^2+^ influx or the cytosolic Ca^2+^ concentration plays a role in the induction of gravitropism, but no direct evidence in support of the Ca^2+^-mediated regulation of gravitropism has been shown to date [69,70]. On the other hand, the hydrotropic response of agravitropic *ageotropum* pea roots was shown to be inhibited by the application of a calcium chelator, ethylene glycol-bis-(β-aminoethyl ether)-N,N,N′,N′-tetraacetic acid (EGTA), or a Ca^2+^ channel blocker [14]. The application of a Ca^2+^ ionophore, A23187, accelerated the hydrotropic response of pea roots or abolished the EGTA-mediated inhibition of hydrotropism [14]. Recently, Shkolnik et al. [15] reported that cytosolic Ca^2+^ acts as a slow and long-distance signal in the hydrotropic response. Upon hydrostimulation, a shootward Ca^2+^ signal travels from the lateral root cap to the elongation zone through the root phloem. This signal becomes distributed asymmetrically upon reaching the elongation zone, where a stronger accumulation of Ca^2+^ is observed at the convex side of the root [15]. The elevation of cytosolic Ca^2+^ and the hydrotropic response were inhibited in roots treated with a chelator (BAPTA-AM), while treatment with a Ca^2+^ ionophore (Br-A23187) enhanced root hydrotropism [15]. MIZ1, an essential molecule for hydrotropism, localizes on the ER surface [71]. Interestingly, the ER-localized Ca^2+^-ATPase pump ECA1 regulates cytosolic Ca^2+^ levels in coordination with MIZ1, and the root curvature is enhanced in the absence of functional ECA1, because of an increase in the cytosolic Ca^2+^ concentration. MIZ1 directly interacts with ECA1 and negatively suppresses ECA1 function, causing an increase in the cytosolic Ca^2+^ levels [15]. Shkolnik et al. [15] consider that an increase in the cytosolic Ca^2+^ level in the lateral root cap is required for hydrotropic bending at the elongation zone. These findings suggest the importance of cytosolic Ca^2+^ in the induction of root hydrotropism.

We also observed a similar transition in the cytosolic Ca^2+^ levels following hydrostimulation in Arabidopsis roots, although it was difficult to detect the asymmetry in the Ca^2+^ distribution in our system (Figure 3). This difference could be due to the non-tagged Yellow Cameleon (YC) 3.6 used in our study, while Shkolnik et al. [15] used the nuclear export signal (NEC)-tagged YC3.6 driven by the UBQ10 promoter for reducing background signals in the cytosol. Nevertheless, in the split-agar assay, the hydrotropic curvature of the Arabidopsis roots became visible 6–12 h after the start of hydrostimulation, reached a peak at 36 h after hydrostimulation, and slightly decreased thereafter (Figure 3A). A reduction in the root curvature at a later stage of the assay was caused by the strengthening of the gravitropic response as the roots deviated from the plumb line in response to a water gradient. The analysis using YC3.6 showed that the cytosolic Ca^2+^ level in the root cap (including the lateral root cap) started increasing at 6–12 h and continued to increase until 36 h after hydrostimulation (Figure 3B,C). The Ca^2+^ increase appeared to expand to other tissues a little later. Relatively, a strong signal of a higher Ca^2+^ level was observed in the lateral root cap. The stele of the elongation zone showed a remarkable increase in cytosolic Ca^2+^ 36–48 h after hydrostimulation (Figure 3B,D). The systemic treatment of Arabidopsis roots with osmotic stress increased the cytosolic Ca^2+^ level in both the root cap and entire root proper (intensively in the stele) as observed in the hydrostimulated roots (Figure 4). Considering that roots asymmetrically perceive osmotic stress in the presence of moisture gradients, the asymmetry of the cytosolic Ca^2+^ level and its relationship with MIZ1 function observed by Shkolnik et al. [15] in hydrostimulated roots could explain an essential part of the signaling system for hydrotropism. However, the sensory cells/tissues that perceive moisture gradients for the induction of the hydrotropic response need to be investigated. The de-tipping of roots with precise and careful surgery or laser ablation of the root tip suppressed gravitropism but did not perturb hydrotropism in Arabidopsis [43]. In rice and cucumber seedlings, de-tipping accelerated the hydrotropic response of roots, probably by inhibiting the gravitropic response [72,73]. It appears that long-distance Ca^2+^ signaling due to hydrostimulation occurs even in de-tipped roots, although a part of the lateral root cap remains intact in some of these roots (Figure 5). We cannot rule out a possibility that hydrostimulation or osmotic stress causes a Ca^2+^ increase in the elongation zone independently of the root cap. Additionally, a previous study showed that MIZ1 functions in the cortex of the elongation zone to induce hydrotropism [43]. A question arises as to how a change in the cytosolic Ca^2+^ concentration in the root cap or root-cap-driven Ca^2+^ signaling is related to MIZ1 function in the cortex. It is therefore important to examine whether the asymmetric perception of water stress in the presence of moisture gradients occurs and results in asymmetrical Ca^2+^ signaling in the root elongation zone itself. Furthermore, the participation of Ca^2+^ channel proteins in the hydrotropic response of Arabidopsis roots was demonstrated in *osca1* mutants, but evidence suggests their importance at the plasma membrane [74]. It is probable that the ER- and MIZ1-mediated regulation of the cytosolic Ca^2+^ concentration is influenced by Ca^2+^ influx from the apoplast.

In addition, Tanaka-Takada et al. [16] showed that plasma membrane-associated cation-binding protein (PCaP1) plays a role in hydrotropism. The N-terminal end of PCaP1 can bind to phosphatidylinositol diphosphates and Ca^2+^ via a CaM complex. PCaP1 binds to phosphatidylinositol diphosphates at low cytosolic Ca^2+^ levels, and phosphatidylinositol diphosphates are replaced with Ca^2+^ at high Ca^2+^ levels. The *pcap1* null mutants exhibited decreased root bending in response to hydrostimulation; however, this phenotype was rescued when *PCaP1* was expressed under the control of an endodermis-specific promoter in the *pcap1* background. PCaP1 likely functions in the endodermis to facilitate the hydrotropic response. Thus, Ca^2+^ signaling could be a crucial player in hydrotropism, although Ca^2+^ signaling across tissues and its relation to MIZ1 regulation in the cortex need to be clarified.

## 5. Crosstalk among ROS, Ca^2+^, and ABA Signaling Pathways

The literature discussed above indicates that ROS, Ca^2+^, and ABA play regulatory roles in the hydrotropic response. However, it is still difficult to propose a model that explains how these three factors interact during hydrotropism, since the dynamics of ROS, Ca^2+^, and ABA have never been tested in a single study. However, thus far, both the ROS-induced Ca^2+^ wave and Ca^2+^-induced ROS wave have been well established as mechanisms for long-distance signal transduction [76]. For example, salt stress sensed at the tip of Arabidopsis roots triggers a Ca^2+^ wave, which is then propagated through the endodermis and cortex, indicating that signal transduction exhibits cell type specificity [77]. It has been demonstrated that Ca^2+^ influx into the cytosol is dependent on the vacuolar ion channel TWO PORE CHANELL1 (TPC1); interestingly, TPC1 is sensitive to H_2_O_2_ [78]. On the other hand, NADPH oxidases are activated by Ca^2+^ either directly via their EF-hand motif or indirectly via CDPKs [79]. The sensitivity of TPC1 to H_2_O_2_ and the induction of NADPH oxidase activity via Ca^2+^ suggests a feed-forward mechanism between ROS and Ca^2+^ that can amplify and transduce the signal through plant organs. The rate of Ca^2+^ waves decreased from 396 µm s^−1^ to 15.5 µm s^−1^ in the *tpc1-2* mutant and to 73 µm s^−1^ in *rbohd* mutants [80]. Hence, ROS wave-dependent systemic signaling depends mostly on vacuolar Ca^2+^ release and the subsequent signal amplification. It would be interesting to test the hydrotropic response of *tpc1-2* mutants to ascertain whether ROS and Ca^2+^ waves play a role in hydrotropic root bending.

Our current state of knowledge, based on the literature, suggests that RBOHs are not involved in the hydrotropic response; on the contrary, the inhibition of RBOHs enhances the rate of root bending. Therefore, RBOH-mediated ROS production is activated during hydrotropism. Is this a side effect of the increase in the cytosolic Ca^2+^ concentration during hydrotropism? Does RBOH activity slow down the hydrotropic response by changing the cell wall structure (stiffening or loosening) as a side effect? These questions deserve further scrutiny.

## 6. Commonalities and Differences between Root Hydrotropism and Leaf Stomatal Movement

It appears that the stomatal and hydrotropic responses share similar mechanisms of stimuli perception and signal transduction. The number of cells (even tissue types) to be coordinated might differ between the two responses, being higher in root hydrotropism; however, ABA is required for both these responses, and ABA perception and signal transduction utilize common circuits (Figure 6). In addition, the release of Ca^2+^ from cellular storage and the involvement of ROS are common to both events. The *rbohd rbohf* double mutant exhibited reduced sensitivity to ABA and consequently less ABA-induced inhibition of root growth than the wild-type [81]. Moreover, these *rbohd* and *rbohf* mutants exhibited defects in ROS generation, an increase in the cytosolic Ca^2+^ concentration, and the activation of Ca^2+^-permeable channels in response to ABA. Treatment with H_2_O_2_ activated the Ca^2+^ current in the mutant roots, indicating the involvement of RBOHs in the cytosolic Ca^2+^ increase. Hence, ROS, Ca^2+^, and ABA are mutually involved in stomatal closure and root hydrotropism. However, the phenomenon of hydrotropism is complex because of the spatial distribution of these events in different root tissues and the amplitude of signals that regulate root growth under an osmotic gradient. Also, the stomatal response occurs after the osmotic regulation of individual cells, whereas the hydrotropic response is caused by the differential growth of a limited number of cells.

## 7. Concluding Remarks

Numerous studies have been conducted on the role of ABA and Ca^2+^ flux in the hydrotropic response of roots. Although the available information suggests that ROS is a negative regulator of the hydrotropic response, its involvement in root hydrotropism with respect to the regulation of the Ca^2+^ and ABA responses remains obscure. The elucidation of the molecular network of ABA, ROS, and Ca^2+^ signaling, together with the functions of MIZ1 and GNOM/MIZ2, will shed a light on root hydrotropism. As discussed above, MIZ1 and ABA function in the cortex; ROS accumulation is observed in the root at the point of the curve; a change in the cytosolic Ca^2+^ level occurs in the root cap and stele of the elongation zone; and the plasma membrane-associated Ca^2+^-binding protein in the endodermis is involved in hydrotropism. The signaling network that controls hydrotropism could be clarified by understanding (1) the roles played by cells/tissues in the function of each regulator, and (2) the relationships among these regulators. To understand the sensory mechanism of hydrostimulation, we may not need to consider a particular apparatus if the asymmetry in the ABA distribution and/or cytosolic Ca^2+^ concentration is established as a result of the differential perception of water stress by the roots in the presence of moisture gradients. However, specific regulators such as a certain type of ion channel or signaling network could play a role in hydrotropism, since MIZ1 and GNOM/MIZ2 are indispensable for hydrotropism in Arabidopsis roots. We did not describe the details here, but it was shown that cytokinin plays an important role in root hydrotropism [82,83]. Interestingly, the asymmetric distribution of cytokinin occurred in hydrotropically stimulated roots and caused differential cell division in the meristematic zone [83]. As discussed in the present paper, the regulation of hydrotropism by cytokinin also needs to be further studied in relation to other regulators responsible for hydrotropism or differential cell expansion in hydrotropically responding roots. Additionally, differences in the regulatory mechanism of hydrotropism among species should be investigated in future studies. We have learned that mechanisms underlying root hydrotropism are quite unique to this process compared with other tropisms such as gravitropism, although root phototropism partially shares the regulatory mechanism with hydrotropism [84]. Answering the questions raised in this review will help to further understand the sensory and signaling mechanisms underlying the hydrotropic response of roots.

## Figures and Tables

**Figure 1 plants-13-01220-f001:**
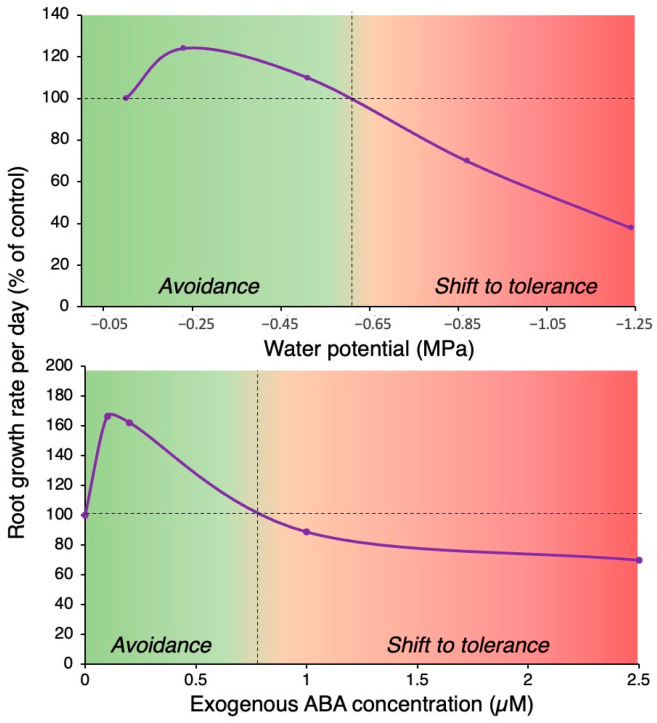
Growth rate of *Arabidopsis thaliana* roots exposed to decreasing water potential and exogenous ABA treatment. Data were taken from van der Weele et al. [33] for water potential and from Li et al. [34] for exogenous ABA. Data presented in the studies were extracted from graphs with WebPlotDigitizer, and graphs in this figure were prepared as relative values to depict avoidance and tolerance responses.

**Figure 2 plants-13-01220-f002:**
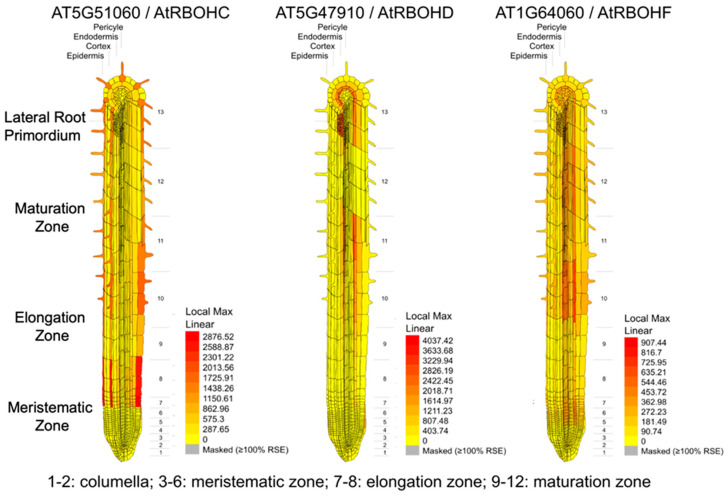
Expression of *RBOHC*, *RBOHD*, and *RBOHF* genes in Arabidopsis roots. Data source: Brady et al. [56] and Winter et al. [57] with eFP Browser.

**Figure 3 plants-13-01220-f003:**
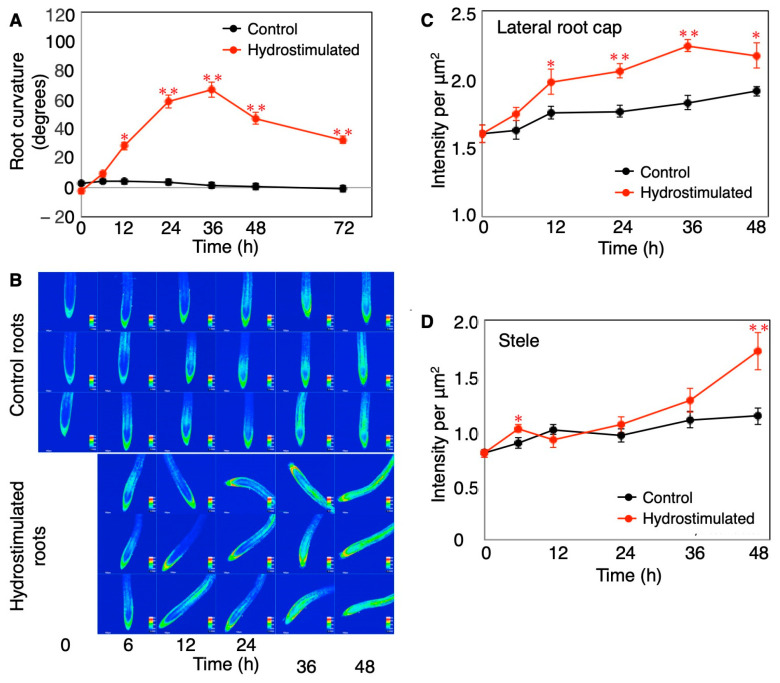
Hydrotropic curvature and cytosolic Ca^2+^ imaging in Arabidopsis roots. (**A**) Time course analysis of hydrotropic response. Roots of three-day-old seedlings of *Arabidopsis thaliana* ecotype Columbia (Col-0) were hydrotropically stimulated using the split-agar assay [75]. Root curvature (degrees) was measured using ImageJ ver. 1.42 and expressed as mean ± SE (*n* = 30). Asterisks indicate significant differences (* *p* < 0.05; ** *p* < 0.01; Student’s *t*-test). (**B**) Time course study of the changes in YC3.6 (Ca^2+^ reporter) signal intensity in control and hydrostimulated roots. Roots were observed under a confocal laser scanning microscope (FV-1000/IX-81, Olympus, Tokyo, Japan), and FRET images were obtained. Three roots from each treatment are shown. (**C**,**D**) Quantification of YC3.6 signal intensity in the lateral root cap (**C**) and stele (**D**) of the elongation zone. Data represent the mean ± SE (*n* = 9–15). Asterisks indicate significant differences (* *p* < 0.05; ** *p* < 0.01; Student’s *t*-test).

**Figure 4 plants-13-01220-f004:**
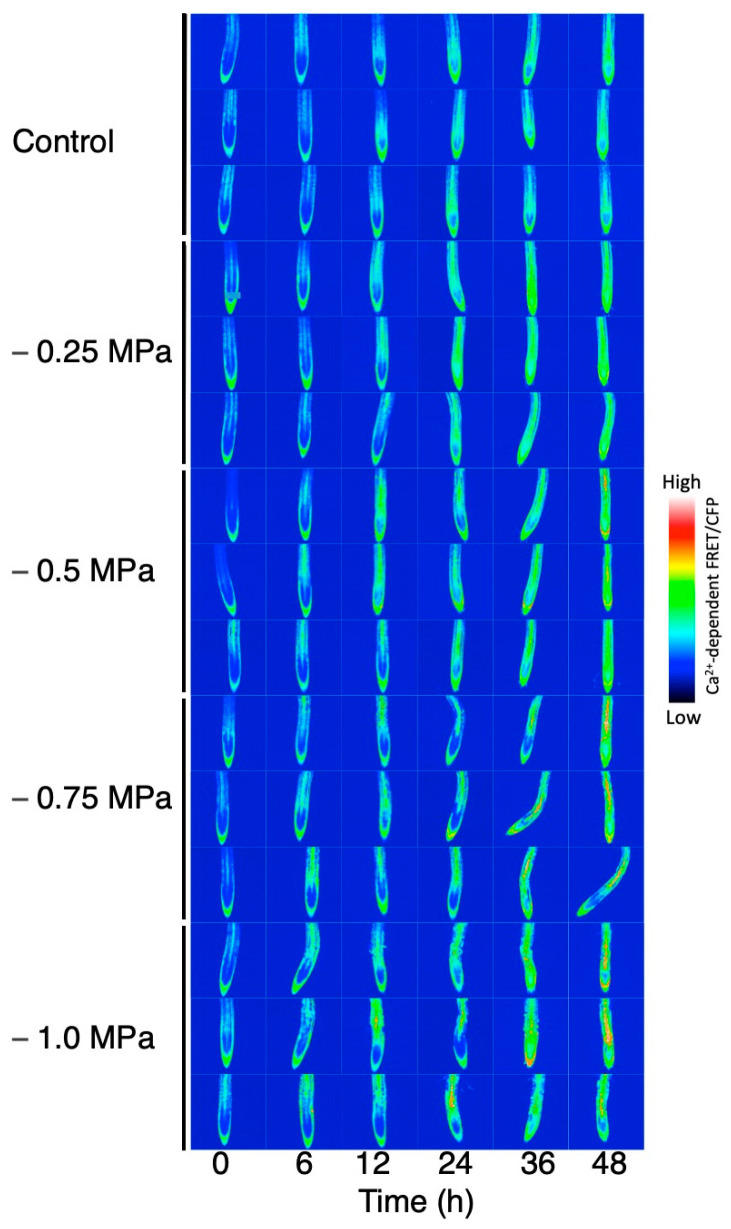
Time course study of the changes in YC3.6 (Ca^2+^ reporter) signal intensity in systemically osmo-stressed roots. Seedling roots were exposed to varying degrees of osmotic stress (0, −0.25, −0.5, −0.75, and −1.0 MPa). Roots were observed under a confocal laser scanning microscope (FV-1000/IX-81, Olympus), and FRET images were obtained. Three roots from each treatment are shown.

**Figure 5 plants-13-01220-f005:**
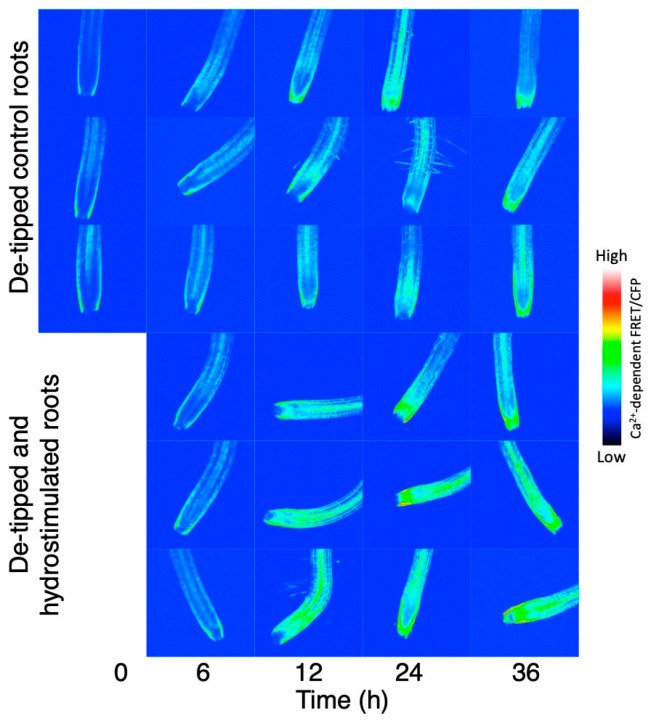
Effects of de-tipping on YC3.6 signal intensity in control and hydrostimulated roots. Root tip (100 μm) was removed with a micro blade under a stereomicroscope (SZX16, Olympus). Roots were observed under a confocal laser scanning microscope (FV-1000/IX-81, Olympus), and FRET images were obtained. Three roots from each treatment are shown.

**Figure 6 plants-13-01220-f006:**
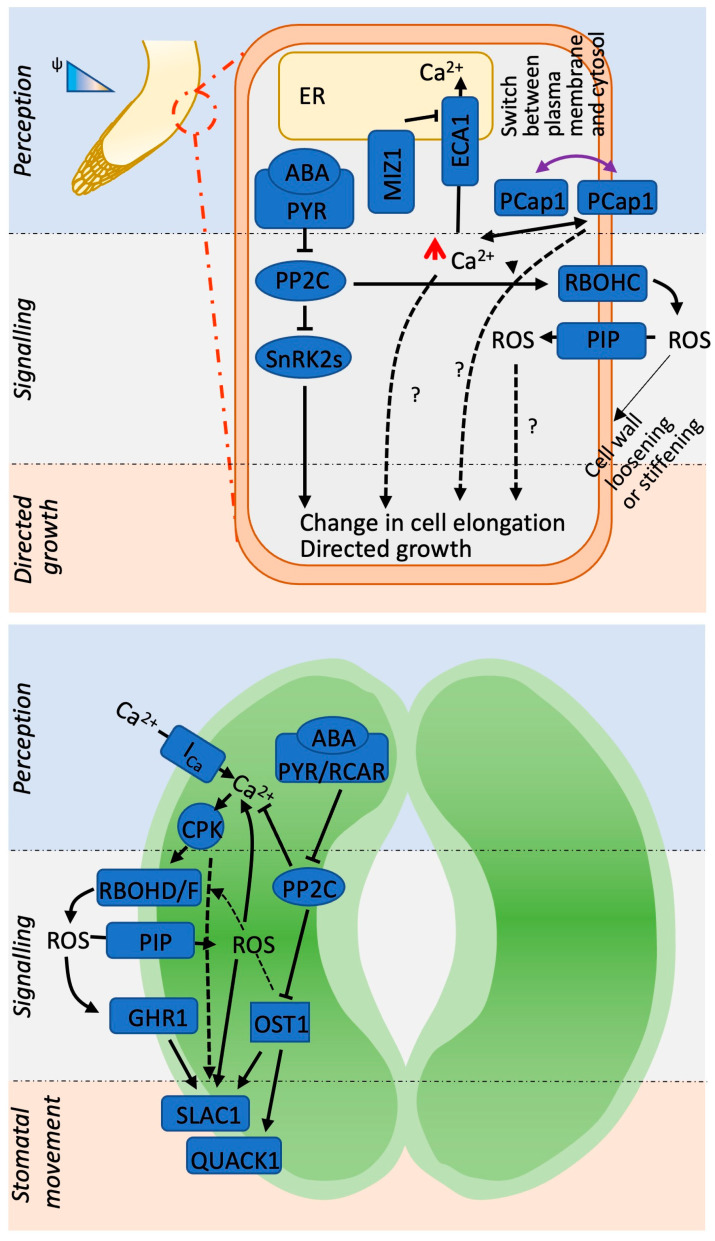
Scheme depicting the commonalities between hydrotropism and stomatal response.

## Data Availability

The data presented in this article are available from the corresponding authors upon request.

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
