# Peer review of "Role of Abscisic Acid, Reactive Oxygen Species, and Ca2+ Signaling in Hydrotropism—Drought Avoidance-Associated Response of Roots"

_plants, 2024, doi:10.3390/plants13091220_

Round 1
Reviewer 1 Report
Comments and Suggestions for Authors
This manuscript appears to be a carefully and well-written. The figures are well documented and demonstrate the points well. I do have a few comments about things that need to be corrected and perhaps explained in a bit more detail.
1) P3, para 1, last line. Delete “however” and move this sentence before the previous sentence (begins with “A question….)
2) P.3 para 3 : Fig1 is this new data for this manuscript? If not what is source?, cite appropriately.
3) P.4, para 2, Line 1 contains an error: abi1-1 is not an ABA biosynthesis mutant (I believe the authors meant aba1-1 as was presumably used in the paper cited.
4) P.5, para 1, line 2-4. Why is “however” used, is not the exogenous application and rescue of aba1-1 consistent with model.. Or does the presumption of the lack of a gradient make it inconsistent. Explain more clearly?
5) P. 5, line 6-8, beginning with “On the other hand” can this be explained more clearly?
6) P.5, para 2, line 20-21, beginning with “however ….” Why, who is we? Explain a bit how it was experimentally determined that it was not differential (what method, tools etc.) Citation also needed here.
7) P.6, para 3 line 7-9, beginning with Therefore …. Logic is not totally clear to me, please
explain more.
8) p.7, para 2, 5 lines from end of para. “The authors (needs citation)…..”
9) p. 8, para 2 , beginning with Tsou …… Is CBP being mentioned in this para, calreticulin?, if calreticulin should be used, not CBP.
10) P.8, para 3, line 3 from end, Delete “The authors” and please insert Scholnik et al if they are the appropriate authors being referred to here.
11) P.8, para4, line 11 and following. The discussion about localization of the calcium levels. It is not clear at all to me (based on data shown in Fig 3 and 5) that the calcium increases are in the root cap, rather they appear to be in the root tip or even just behind the root tip. That is certainly substantiated by hydrotropism being sustained when the cap (and perhaps some of tip) is removed. Also the sentences starting with “Systemic treatment …….) with osmotic stress increased cytosolic Calcium levels [insert in the stele] as observed ….
12) Figure 6, why not place calcium in this model for hydrotropism. (at least with some ??)
13) P. 12 para 2, line 8; No new data shown in this manuscript convinces me that the Ca changes are in the root cap rather than the root tip. Unless there are other publications that support that (and are cited), this statement seems to me to be too speculative.
Reviewer 2 Report
Comments and Suggestions for Authors
Drouugh is a very important limiting factor for crop yield in the world. Therefore, elucidation of drought resistantce mechanism is meaningful both in basic and applied field. In this review paper, authors summerized root hydrotropism phenomenon, which are regulated by ABA, Ca2+, and ROS. The manuscript is well written and organized. Because of diffficulty on the study of root, there were very slow progresses on this topic. However, I found that there are some reports on this topic recently, therefore, my only suggestion is adding some new information on this topic using recent findings.
Author Response
To Reviewer 2.
Authors: Thanks for the reviewer’s evaluation and suggestion!
We are not sure what information or publication the reviewer meant, but we believe that information (references) important for reviewing root hydrotropism in relation to the roles for ABA, ROS, and Ca2+ signaling are sufficiently included and discussed.
Reviewer 3 Report
Comments and Suggestions for Authors
The review article "Regulatory Mechanisms and Signaling Pathways in Plant Root Hydrotropism: Insights from Drought Stress Responses" by Uzilday et al., shows an overview of the regulatory mechanisms and signaling pathways involved in plant root hydrotropism, with an emphasis on novel regulators such as MIZ1 and GNOM/MIZ2, and signaling molecules including abscisic acid (ABA), reactive oxygen species (ROS), and calcium ions (Ca2+).
The abstract provides a succinct summary of the key findings, highlighting the involvement of ABA, ROS, and Ca2+ signaling not only in root hydrotropism but also in plant responses to drought stress. It elucidates how these signaling molecules act in concert within specific regions of the root, such as the cortex of the elongation zone, root curvature, and entire root tip, to mediate hydrotropic responses.
The authors appropriately acknowledge the need for further research to unravel the complexities of hydrotropism regulation, including a detailed analysis of tissue-specific functions of regulators and their signaling networks. By integrating insights from studies on drought stress responses, the review offers a valuable synthesis of recent findings and raises pertinent questions for future investigation.
Overall, the review is sound, interesting, and in my opinion can be accepted after the follow minor adjustment:
- Present the figures in larger size and resolution, especially increasing the font size so that the text and indications in the diagrams are more visible.
Author Response
To Reviewer 3.
Authors: Thanks for the reviewer’s evaluation and suggestion!
We expect that figures will be enlarged and adjusted by editorial work using margins around the figures. If it is a matter of font size we used, we can amend it, of course. We leave it to editorial judgment.